# Coreference in Wikipedia: Main Concept Resolution

## Abstract

Wikipedia is a resource of choice exploited in many NLP applications, yet we are not aware of recent attempts to adapt coreference resolution to this resource. In this work, we revisit a seldom studied task which consists in identifying in a Wikipedia article all the mentions of the main concept being described. We show that by exploiting the Wikipedia markup of a document, as well as links to external knowledge bases such as Freebase, we can acquire useful information on entities that helps to classify mentions as coreferent or not. We designed a classifier which drastically outperforms fair baselines built on top of state-of-the-art coreference resolution systems. We also measure the benefits of this classifier in a full coreference resolution pipeline applied to Wikipedia texts.

## 1 Introduction

Coreference Resolution (CR) is the task of identifying all mentions of entities in a document and grouping them into equivalence classes. CR is a prerequisite for many NLP tasks. For example, in Open Information Extraction (OIE) (Yates et al., 2007), one acquires subject-predicate-object relations, many of which (e.g., <the foundation stone, was laid by, the Queen s daughter>) are useless because the subject or the object contains material coreferring to other mentions in the text being mined.

Most CR systems, including state-of-the-art ones (Durrett and Klein, 2014; Martschat and Strube, 2015; Clark and Manning, 2015) are essentially adapted to news-like texts. This is basically imputable to the availability of large datasets where this text genre is dominant. This includes resources developed within the Message Understanding Conferences (e.g., (Hirshman and Chinchor, 1998)) or the Automatic Content Extraction (ACE) program (e.g., (Doddington et al., 2004)), as well as resources developed within the collaborative annotation project OntoNotes (Pradhan et al., 2007).

It is now widely accepted that coreference resolution systems trained on newswire data performs poorly when tested on other text genres (Hendrickx and Hoste, 2009; Schäfer et al., 2012), including Wikipedia texts, as we shall see in our experiments.

Wikipedia is a large, multilingual, highly structured, multi-domain encyclopedia, providing an increasingly large wealth of knowledge. It is known to contain well-formed, grammatical and meaningful sentences, compared to say, ordinary internet documents. It is therefore a resource of choice in many NLP systems, see (Medelyan et al., 2009) for a review of some pioneering works.

While being a ubiquitous resource in the NLP community, we are not aware of much work conducted to adapt CR to this text genre. Two notable exceptions are (Nguyen et al., 2007) and (Nakayama, 2008), two studies dedicated to extract tuples from Wikipedia articles. Both studies demonstrate that the design of a dedicated rule-based CR system leads to improved extraction accuracy. The focus of those studies being information extraction, the authors did not spend much efforts in designing a fully-fledged CR designed for Wikipedia, neither did they evaluate it on a coreference resolution task.

Our main contribution in this work is to revisit the task initially discussed in (Nakayama, 2008) which consists in identifying in a Wikipedia article all the mentions of the concept being described by this article. We refer to this concept as the "main concept" (MC) henceforth. For instance, within

the article `Chilly_Gonzales`, the task is to find all proper (e.g. *Gonzales*, *Beck*), nominal (e.g. *the performer*) and pronominal (e.g. *he*) mentions that refer to the MC "Chilly Gonzales".

For us, revisiting this task means that we propose a testbed for evaluating systems designed for it, and we compare a number of state-of-the-art systems on this testbed. More specifically, we frame this task as a binary classification problem, where one has to decide whether a detected mention refers to the MC. Our classifier exploits carefully designed features extracted from Wikipedia markup and characteristics, as well as from Freebase; many of which we borrowed from the related literature.

We show that our approach outperforms state-of-the-art generic coreference resolution engines on this task. We further demonstrate that the integration of our classifier into the state-of-the-art rule-based coreference system of Lee et al. (2013) improves the detection of coreference chains in Wikipedia articles.

The paper is organized as follows. We discuss related works in Section 2. We describe in Section 3 the baselines we built on top of two state-of-the-art coreference resolution systems, and present our approach in Section 4. We describe the dataset we exploited in Section 5. We explain experiments we conducted on a Wikipedia dataset in section 6, and conclude in Section 7.

## 2 Related Works

Our approach is inspired by, and extends, previous works on coreference resolution which show that incorporating external knowledge into a CR system is beneficial. In particular, a variety of approaches (Ponzetto and Strube, 2006; Ng, 2007; Haghighi and Klein, 2009) have been shown to benefit from using external resources such as Wikipedia, WordNet (Miller, 1995), or YAGO (Suchanek et al., 2007). Ratinov and Roth (2012) and Hajishirzi et al. (2013) both investigate the integration of named-entity linking into machine learning and rule-based coreference resolution system respectively. They both use GLOW (Ratinov et al., 2011) a *wikification* system which associates detected mentions with their equivalent entity in Wikipedia. In addition, they assign to each mention a set of highly accurate knowledge attributes extracted from Wikipedia and Freebase (Bollacker et al., 2008), such as the Wikipedia categories, gender, nationality, aliases, and NER type (ORG, PER, LOC, FAC, MISC).

One issue with all the aforementioned studies is that inaccuracies often cause cascading errors in the pipeline (Zheng et al., 2013). Consequently, most authors concentrate on high-precision linking at the cost of low recall.

Dealing specifically with Wikipedia articles, we can directly exploit the wealth of markup available (redirects, internal links, links to Freebase) without resorting to named-entity linking, thus benefiting from much less ambiguous information on mentions.

## 3 Baselines

Since there is no system readily available for our task, we devised four baselines on top of two available coreference resolution systems. Given the output of a CR system applied on a Wikipedia article, our goal here is to isolate the coreference chain that represents the main concept. We experimented with several heuristics, yielding the following baselines.

**B1** picks the longest coreference chain identified and considers that its mentions are those that co-refer to the main concept. The underlying assumption is that the most mentioned concept in a Wikipedia article is the main concept itself.

**B2** picks the longest coreference chain identified if it contains a mention that exactly matches the MC title, otherwise it checks in decreasing order (longest to shortest) for a chain containing the title. We expect this baseline to be more precise than the previous one overall.

It turns out that, for both CR systems, mentions of the MC often are spread over several coreference chains. Therefore we devised two more baselines that aggregate chains, with an expected increase in recall.

**B3** conservatively aggregates chains containing a mention that exactly matches the MC title.

**B4** more loosely aggregates all chains that contain at least one mention whose span is a substring of the title.[1] For instance, given the main concept *Barack Obama*, we concatenate all chains containing either *Obama* or *Barack*

---

[1] Grammatical words are not considered for matching.

in their mentions. Obviously, this baseline should show a higher recall than the previous ones, but risks aggregating mentions that are not related to the MC. For instance, it will aggregate the coreference chain referring to *University of Sydney* concept with a chain containing the mention *Sydney*.

We observed that, for pronominal mentions, those baselines were not performing very well in terms of recall. With the aim of increasing recall, we added to the chain all the occurrences of pronouns found to refer to the MC (at least once) by the baseline. This heuristic was first proposed by Nguyen et al. (2007). For instance, if the pronoun *he* is found in the chain identified by the baseline, all pronouns *he* in the article are considered to be mentions of the MC *Barack Obama*. Obviously, there are cases where those pronouns do not co-refer to the MC, but this step significantly improves the performance on pronouns.

## 4 Approach

Our approach is composed of a preprocessor which computes a representation of each mention in an article as well as its main concept; and a feature extractor which compares both representations for inducing a set of features.

### 4.1 Preprocessing

We extract mentions using the same mention detection algorithm embedded in `Dcoref` and `Scoref`. This algorithm described in (Raghunathan et al., 2010) extracts all named-entities, noun phrases and pronouns, and then removes spurious mentions.

We leverage the hyperlink structure of the article in order to enrich the list of mentions with shallow semantic attributes. For each link found within the article under consideration, we look through the candidate list for all mentions that match the surface string of the link. We assign to those mentions the attributes (entity type, gender and number) extracted from the Freebase entry (if it exists) corresponding to the Wikipedia article the hyperlink points to. This module behaves as a substitute to the named-entity linking pipelines used in other works, such as (Ratinov and Roth, 2012; Hajishirzi et al., 2013). We expect it to be of high quality because it exploits human-made links. We use the `WikipediaMiner` (Milne and Witten, 2008) API for easily accessing any piece

of structure (clean text, labels, internal links, redirects, etc) in Wikipedia, and Jena[2] to index and query Freebase.

---

**string span**
> *San Fernando Valley region
> of the city of Los Angeles*

**head word span**
> *region*

**span up to the head noun**
> *San Fernando Valley region*

**coarse attribute**
> $\emptyset$, *neutral, singular*

---

Figure 1: Representation of a mention.

In the end, we represent a mention by three strings (actual mention span, head word, and span up to the head noun), as well as its coarse attributes (entity type, gender and number). Figure 1 shows the representation collected for the mention *San Fernando Valley region of the city of Los Angeles* found in the `Los_Angeles_Pierce_College` article.

---

**title** (W)
> *Los Angeles Pierce College*

**inferred type** (W)
*Los Angeles Pierce College, also known as Pierce College and just Pierce, is a community college that serves . . .*
> *college*

**name variants** (W,F)
> *Pierce Junior College*, LAPC

**entity type** (F)
> College/University

**coarse attributes** (F)
> ORG, neutral, singular

---

Figure 2: Representation of a Wikipedia concept. The source from which the information is extracted is indicated in parentheses: (W)ikipedia, (F)reebase.

We represent the main concept of a Wikipedia article by its **title**, its **inferred type** (a common noun inferred from the first sentence of the article). Those attributes were used by Nguyen et al. (2007) to heuristically link a mention to the main concept of an article. We further extend this representation by the MC **name variants** extracted

---

[2] http://jena.apache.org

from the markup of Wikipedia (redirects, text anchored in links) as well as aliases from Freebase; the MC **entity types** we extracted from the Freebase `notable types` attribute, and its **coarse attributes** extracted from Freebase, such as its NER type, its gender and number. If the concept category is a person (PER), we import the `profession` attribute. Figure 2 illustrates the information we collect for the Wikipedia concept `Los_Angeles_Pierce_College`.

### 4.2 Feature Extraction

We experimented with a few hundred features for characterizing each mention, focusing on the most promising ones that we found simple enough to compute. In part, our features are inspired by coreference systems that use Wikipedia and Freebase as feature sources (see Section 2). These features, along with others related to the characteristics of Wikipedia texts, allow us to recognize mentions of the MC more accurately than current CR systems. We make a distinction between features computed for pronominal mentions and features computed from the other mentions.

#### 4.2.1 Non-pronominal Mentions

For each mention, we compute seven families of features we sketch below.

**base** Number of occurrences of the mention span and the mention head found in the list of candidate mentions. We also add a normalized version of those counts (frequency / total number of mentions).

**title, inferred type, name variants, entity type** Most often, a concept is referred to by its name, one of its variants, or its type which are encoded in the four first fields of our MC representation. We define four families of comparison features, each corresponding to one of the first four fields of a MC representation (see Figure 2). For instance, for the title family, we compare the title text span with each of the text spans of the mention representation (see Figure 1). A comparison between a field of the MC representation and a mention text span yields 10 boolean features. These features encode string similarities (exact match, partial match, one being the substring of another, sharing of a number of words, etc.). An eleventh feature is the semantic relatedness score of Wu and

Palmer (1994). For **title**, we therefore end up with 3 sets of 11 feature vectors.

**tag** Part-of-speech tags of the first and last words of the mention, as well as the tag of the words immediately before and after the mention in the article. We convert this into $34 \times 4$ binary features (presence/absence of a specific combination of tags).

**main** Boolean features encoding whether the MC and the mention **coarse attributes** matches; also we use conjunctions of all pairs of features in this family.

#### 4.2.2 Pronominal Mentions

We characterize pronominal mentions by five families of features, which, with the exception of the first one, all capture information extracted from Wikipedia.

**base** The pronoun span itself, number, gender and person attributes, to which we add the number of occurrences of the pronoun, as well as its normalized count. The most frequently occurring pronoun in an article is likely to co-refer to the main concept, and we expect these features to capture this to some extent.

**main** MC coarse attributes, such as NER type, gender, number (see Figure 2).

**tag** Part-of-speech of the previous and following tokens, as well as the previous and the next POS bigrams (this is converted into 2380 binary features).

**position** Often, pronouns at the beginning of a new section or paragraph refer to the main concept. Therefore, we compute 5 (binary) features encoding the relative position (first, first tier, second tier, last tier, last) of a mention in the sentence, paragraph, section and article.

**distance** Within a sentence, we search before and after the mention for an entity that is compatible (according to Freebase information) with the pronominal mention of interest. If a match is found, one feature encodes the distance between the match and the mention; another feature encodes the number of other compatible pronouns in the same sentence. We expect that this family of features will

help the model to capture the presence of local (within a sentence) co-references.

## 5 Dataset

As our approach is dedicated to Wikipedia articles, we used a dedicated resource we call WCR[3], and whose details will be described in the non-anonymized version of this paper. It consists of 30 documents, comprising 60k tokens annotated with the OntoNotes project guidelines (Pradhan et al., 2007). Each mention is annotated with three attributes: the mention type (named-entity, noun phrase, or pronominal), the coreference type (identity, attributive or copular) and the equivalent Freebase entity if it exists. The resource contains roughly 7000 non singleton mentions, among which 1800 refer to the main concept, which is to say that 30 chains out of 1469 make up for 25% of the mentions annotated.

Since most coreference resolution systems for English are trained and tested on ACE (Doddington et al., 2004) or OntoNotes (Hovy et al., 2006) resources, it is interesting to measure how state-of-the art systems perform on the WCR dataset. To this end, we ran a number of recent CR systems: the rule-based system of (Lee et al., 2013) we call it `Dcoref`; the Berkeley systems described in (Durrett and Klein, 2013; Durrett and Klein, 2014); the latent model of Martschat and Strube (2015) we call it `Cort` in Table 1; and the system described in (Clark and Manning, 2015) we call it `Scoref` which achieved the best results to date on the CoNLL 2012 test set.

| System | WCR | OntoNotes |
|---|---|---|
| Dcoref | 51.77 | 55.59 |
| Durrett and Klein (2013) | 51.01 | 61.41 |
| Durrett and Klein (2014) | 49.52 | 61.79 |
| Cort | 49.94 | 62.47 |
| Scoref | 46.39 | 63.61 |

Table 1: CoNLL F1 score of recent state of the art systems on the WCR dataset, and the 2012 OntoNotes test data for predicted mentions.

We evaluate the systems on the whole dataset, using the v8.01 of the CoNLL scorer[4] (Pradhan et al., 2014). The results are reported in Table 1 along with the performance of the systems on the

---

[3]http://www.anonymized.org
[4]http://conll.github.io/reference-coreference-scorers

CoNLL 2012 test data (Pradhan et al., 2012). Expectedly, the performance of all systems dramatically decrease on WCR, which calls for further research on adapting the coreference resolution technology to new text genres. What is more surprising is that the rule-based system of (Lee et al., 2013) works better than the machine-learning based systems on the WCR dataset. Also, the ranking of the statistical systems on this dataset differs from the one obtained on the OntoNotes test set.

The WCR dataset is far smaller than the OntoNotes one; still, we paid attention to sample Wikipedia articles of various characteristics: size, topic (people, organizations, locations, events, etc.) and internal link density. Therefore, we believe our results to be representative. Those results further confirm the conclusions in (Hendrickx and Hoste, 2009), which show that a CR system trained on news-paper significantly under-performs on data coming from users comments and blogs. Nevertheless, statistical systems can be trained or adapted to the WCR dataset, a point we leave for future investigations.

We generated baselines for all the systems discussed in this section, but found results derived from statistical approaches to be close enough that we only include results of two systems in the sequel: `Dcoref` (Lee et al., 2013) and `Scoref` (Clark and Manning, 2015). We choose these two because they use the same pipeline (parser, mention detection, etc), while applying very different techniques (rules versus machine learning).

## 6 Experiments

In this section, we first describe the data preparation we conducted (section 6.1), and provide details on the classifier we trained (section 6.2). Then, we report experiments we carried out on the task of identifying the mentions co-referent (positive class) to the main concept of an article (section 6.3). We compare our approach to the baselines described in section 3, and analyze the impact of the families of features described in section 4. We also investigate a simple extension of `Dcoref` which takes advantage of our classifier for improving coreference resolution (section 6.4).

### 6.1 Data Preparation

Each article in WCR was part-of-speech tagged, syntactically parsed and the named-entities were

identified. This was done thanks to the `Stanford CoreNLP` toolkit (Manning et al., 2014). Since WCR does not contain singleton mentions (in conformance to the OntoNotes guidelines), we consider the union of WCR mentions and all mentions predicted by the method described in (Raghunathan et al., 2010). Overall, we added about 13 400 automatically extracted mentions (singletons) to the 7 000 coreferent mentions annotated in WCR. In the end, our training set consists of 20 362 mentions: 1 334 pronominal ones (627 of them referring to the MC), and 19 028 non-pronominal ones (16% of them referring to the MC).

## 6.2 Classifier

We trained two Support Vector Machine classifiers (Cortes and Vapnik, 1995), one for pronominal mentions and one for non-pronominal ones, making use of the LIBSVM library (Chang and Lin, 2011) and the features described in Section 4.2. For both models, we selected[5] the C-support vector classification and used a linear kernel. Since our dataset is unbalanced (at least for non-pronominal mentions), we penalized the negative class with a weight of 2.0.

During training, we do not use gold mention attributes, but we automatically enrich mentions with the information extracted from Wikipedia and Freebase, as described in Section 4.

## 6.3 Main Concept Resolution Performance

We focus on the task of identifying all the mentions referring to the main concept of an article. We measure the performance of the systems we devised by average precision, recall and F1 rates computed by a 10-fold cross-validation procedure.

The results of the baselines and our approach are reported in Table 4. Clearly, our approach outperforms all baselines for both pronominal and non-pronominal mentions, and across all metrics. On all mentions, our best classifier yields an absolute F1 increase of 13 points over `Dcoref`, and 15 points over `Scoref`.

In order to understand the impact of each family of features we considered in this study, we trained various classifiers in a greedy fashion. We started with the simplest feature set (**base**) and gradually added one family of features at a time, keeping

---

[5]We tried with less success other configurations on a held-out dataset.

at each iteration the one leading to the highest increase in F1. The outcome of this process for the pronominal mentions is reported in Table 2.

A baseline that always considers that a pronominal mention is co-referent to the main concept results in an F1 measure of 63.7%. This naive baseline is outperformed by the simplest of our model (**base**) by a large margin (over 10 absolute points). We observe that recall significantly improves when those features are augmented with the MC coarse attributes (**+main**). In fact, this variant already outperforms all the `Dcoref`-based baselines in terms of F1 score. Each feature family added further improves the performance overall, leading to better precision and recall than any of the baselines tested.

| | P | R | F1 |
|---|---|---|---|
| always positive | 46.70 | 100.00 | 63.70 |
| **base** | 70.34 | 78.31 | 74.11 |
| **+main** | 74.15 | 90.11 | 81.35 |
| **+position** | 80.43 | 89.15 | 84.57 |
| **+tag** | 82.12 | 90.11 | 85.93 |
| **+distance** | 85.46 | 92.82 | 88.99 |

Table 2: Performance of our approach on the pronominal mentions, as a function of the features.

Inspection shows that most of the errors on pronominal mentions are introduced by the lack of information on noun phrase mentions surrounding the pronouns. In example (f) shown in Figure 3, the classifier associates the mention *it* with the MC instead of *the Johnston Atoll " Safeguard C " mission*.

| | P | R | F1 |
|---|---|---|---|
| **base** | 60.89 | 62.24 | 61.56 |
| **+title** | 85.56 | 68.03 | 75.79 |
| **+inferred type** | 87.45 | 75.26 | 80.90 |
| **+name variants** | 86.49 | 81.12 | 83.72 |
| **+entity type** | 86.37 | 82.99 | 84.65 |
| **+tag** | 87.09 | 85.46 | 86.27 |
| **+main** | 91.65 | 85.88 | 88.67 |

Table 3: Performance of our approach on the non-pronominal mentions, as a function of the features.

Table 3 reports the results obtained for the non-pronominal mentions classifier. The simplest classifier is outperformed by most baselines in terms of F1. Still, this model is able to correctly match mentions in example (a) and (b) of Figure 3 simply

|  | Pronominal | | | Non Pronominal | | | All | | |
|---|---|---|---|---|---|---|---|---|---|
|  | P | R | F1 | P | R | F1 | P | R | F1 |
| Dcoref | | | | | | | | | |
| B1 | 64.51 | 76.55 | 70.02 | 70.33 | 63.09 | 66.51 | 67.92 | 67.77 | 67.85 |
| B2 | 76.45 | 50.23 | 60.63 | 83.52 | 49.57 | 62.21 | 80.90 | 49.80 | 61.65 |
| B3 | 76.39 | 65.55 | 70.55 | 83.67 | 56.20 | 67.24 | 80.72 | 59.45 | 68.47 |
| B4 | 71.74 | 83.41 | 77.13 | 74.39 | 75.59 | 74.98 | 73.30 | 78.31 | 75.77 |
| | | | | | | | | | |
| Scoref | | | | | | | | | |
| B1 | 76.59 | 78.30 | 77.44 | 54.66 | 39.37 | 45.77 | 64.11 | 52.91 | 57.97 |
| B2 | 89.59 | 74.16 | 81.15 | 69.90 | 31.20 | 43.15 | 79.69 | 46.14 | 58.44 |
| B3 | 83.91 | 77.35 | 80.49 | 73.17 | 55.44 | 63.08 | 77.39 | 63.06 | 69.49 |
| B4 | 78.48 | 90.74 | 84.17 | 67.51 | 67.85 | 67.68 | 71.68 | 75.81 | 73.69 |
| | | | | | | | | | |
| this work | 85.46 | 92.82 | 88.99 | 91.65 | 85.88 | 88.67 | 89.29 | 88.30 | 88.79 |

Table 4: Performance of the baselines on the task of identifying all MC coreferent mentions.

because those mentions are frequent within their respective article. Of course, such a simple model is often wrong as in example (c), where all mentions *the United States* are associated to the MC, simply because this is a frequent mention.

a MC= *Anatole France*
*France* is also widely believed to be the model for narrator Marcel's literary idol Bergotte in Marcel Proust's In Search of Lost Time.

b MC= *Harry Potter and the Chamber of Secrets*
Although Rowling found it difficult to finish *the book*, it won . . . .

c MC= *Barack Obama*
On August 31, 2010, Obama announced that *the United States*\* combat mission in Iraq was over.

d MC= *Houston Texans*
In 2002, *the team* wore a patch commemorating their inaugural season...

e MC= *Houston Texans*
The name Houston Oilers was unavailable to *the expansion team...*

f MC= *Johnston Atoll*
In 1993 , Congress appropriated no funds for the Johnston Atoll Safeguard C mission , bringing *it*\* to an end.

g MC= *Houston Texans*
The Houston Texans are a professional American football team based in *Houston*\* , Texas.

Figure 3: Examples of mentions (underlined) associated with the MC. An asterisk indicates wrong decisions.

The **title** feature family drastically increases precision, and the resulting classifier (+**title**) outperforms all the baselines in terms of F1 score. Adding the **inferred type** feature family gives a further boost in recall (7 absolute points) with no loss in precision (gain of almost 2 points). For instance, the resulting classifier can link the mention *the team* to the MC *Houston Texans* (see example (d)) because it correctly identifies the term *team* as a type. The family **name variants** also gives a nice boost in recall, in a slight expense of precision. This drop is due to some noisy redirects in Wikipedia, misleading our classifier. For instance, *Johnston and Sand Islands* is a redirect of the `Johnston_Atoll` article. The **entity type** family further improves performance, mainly because it plays a role similar to the **inferred type** features extracted from Freebase. This indicates that the noun type induced directly from the first sentence of a Wikipedia article is pertinent and can complement the types extracted from Freebase when available or serve as proxy when they are missing. Finally, the **main** family significantly increases precision (over 4 absolute points) with no loss in recall. To illustrate a negative example, the resulting classifier wrongly recognizes mentions referring to the town *Houston* as coreferent to the football team in example (g). We handpicked a number of classification errors and found that most of these are difficult coreference cases. For instance, our best classifier fails to recognize that the mention *the expansion team* refers to the main concept *Houston Texans* in example (e).

| System | MUC | | | B³ | | | CEAF$\phi_4$ | | | CoNLL |
|---|---|---|---|---|---|---|---|---|---|---|
| | P | R | F1 | P | R | F1 | P | R | F1 | F1 |
| Dcoref | 61.59 | 60.42 | 61.00 | 53.55 | 43.33 | 47.90 | 42.68 | 50.86 | 46.41 | 51.77 |
| D&K (2013) | 68.52 | 55.96 | 61.61 | 59.08 | 39.72 | 47.51 | 48.06 | 40.44 | 43.92 | 51.01 |
| D&K (2014) | 63.79 | 57.07 | 60.24 | 52.55 | 40.75 | 45.90 | 45.44 | 39.80 | 42.43 | 49.52 |
| M&S (2015) | 70.39 | 53.63 | 60.88 | 60.81 | 37.58 | 46.45 | 47.88 | 38.18 | 42.48 | 49.94 |
| C&M (2015) | 69.45 | 49.53 | 57.83 | 57.99 | 34.42 | 43.20 | 46.61 | 33.09 | 38.70 | 46.58 |
| Dcoref++ | 66.06 | 62.93 | 64.46 | 57.73 | 48.58 | 52.76 | 46.76 | 49.54 | 48.11 | 55.11 |

Table 5: Performance of `Dcoref++` on WCR compared to state of the art systems, including in order: Lee et al. (2013); Durrett and Klein (2013) - Final; Durrett and Klein (2014) - Joint; Martschat and Strube (2015) - Ranking:Latent; Clark and Manning (2015) - Statistical mode with clustering.

## 6.4 Coreference Resolution Performance

Identifying all the mentions of the `MC` in a Wikipedia article is certainly useful in a number of NLP tasks (Nguyen et al., 2007; Nakayama, 2008). Finding all coreference chains in a Wikipedia article is worth studying. In the following, we describe an experiment where we introduced in `Dcoref` a new high-precision sieve which uses our classifier[6]. Sieves in `Dcoref` are ranked in decreasing order of precision, and we ranked this new sieve first. The aim of this sieve is to construct the coreference chain equivalent to the main concept. It merges two chains whenever they both contain mentions to the `MC` according to our classifier. We further prevent other sieves from appending new mentions to the `MC` coreference chain.

We ran this modified system (called `Dcoref++`) on the WCR dataset, where mentions were automatically predicted. The results of this system are reported in Table 5, measured in terms of MUC (Vilain et al., 1995), B3 (Bagga and Baldwin, 1998), CEAF$\phi_4$ (Luo, 2005) and the average F1 CoNLL score (Denis and Baldridge, 2009).

We observe an improvement for `Dcoref++` over the other systems, for all the metrics. In particular, `Dcoref++` increases by 4 absolute points the CoNLL F1 score. This shows that early decisions taken by our classifier benefit other sieves as well. It must be noted, however, that the overall gain in precision is larger than the one in recall.

## 7 Conclusion

We developed a simple yet powerful approach that accurately identifies all the mentions that co-refer

---

[6]We use predicted results from 10-fold cross-validation.

to the concept being described in a Wikipedia article. We tackle the problem with two (pronominal and non-pronominal) models based on well designed features. The resulting system is compared to baselines built on top of state-of-the-art systems adapted to this task. Despite being relatively simple, our model reaches 89 % in F1 score, an absolute gain of 13 F1 points over the best baseline. We further show that incorporating our system into the Stanford deterministic rule-based system (Lee et al., 2013) leads to an improvement of 4% in F1 score on a fully fledged coreference task.

In order to allow other researchers to reproduce our results, and report on new ones, we share all the datasets we used in this study. We also provide a dump of all the mentions in English Wikipedia our classifier identified as referring to the main concept, along with information we extracted from Wikipedia and Freebase. This will be available at `www.somewhere.country`.

A natural extension of this work is to identify all coreference relations in a Wikipedia article, a task we are currently investigating.

## Acknowledgments

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
