# Peer review of "Coreference in Wikipedia: Main Concept Resolution"

_CoNLL 2016 — decision unknown_

[Official Review · Reviewer 1 · rating 5 · confidence 4]
soundness 5 · originality 4 · clarity 3 · impact 4 · substance 4 · appropriateness 5 · meaningful comparison 4 · replicability 4 · presentation format Oral Presentation

The authors present a new version of the coreference task tailored to
Wikipedia. The task is to identify the coreference chain specifically
corresponding to the entity that the Wikipedia article is about.  The authors
annotate 30 documents with all coreference chains, of which roughly 25% of the
mentions refer to the "main concept" of the article. They then describe some
simple baselines and a basic classifier which outperforms these. Moreover, they
integrate their classifier into the Stanford (rule-based) coreference system
and see substantial benefit over all state-of-the-art systems on Wikipedia.

I think this paper proposes an interesting twist on coreference that makes good
sense from an information extraction perspective, has the potential to somewhat
revitalize and shake up coreference research, and might bridge the gap in an
interesting way between coreference literature and entity linking literature. 
I am sometimes unimpressed by papers that dredge up a new task that standard
systems perform poorly on and then propose a tweak so that their system does
better. However, in this case, the actual task itself is quite motivating to me
and rather than the authors fishing for a new domain to run things in, it
really does feel like "hey, wait, these standard systems perform poorly in a
setting that's actually pretty important."

THE TASK: Main concept resolution is an intriguing task from an IE perspective.
 I can imagine many times where documents revolve primarily around a particular
entity (biographical documents, dossiers or briefings about a person or event,
clinical records, etc.) and where the information we care about extracting is
specific to that entity. The standard coreference task has always had the issue
of large numbers of mentions that would seemingly be pretty irrelevant for most
IE problems (like generic mentions), and this task is unquestionably composed
of mentions that actually do matter.

From a methodology standpoint, the notion of a "main concept" provides a bit of
a discourse anchor that is useful for coreference, but there appears to still
be substantial overhead to improve beyond the baselines, particularly on
non-pronominal mentions. Doing coreference directly on Wikipedia also opens the
doors for more interesting use of knowledge, which the authors illustrate here.
So I think this domain is likely to be an interesting testbed for ideas which
would improve coreference overall, but which in the general setting would be
more difficult to get robust improvements with and which would be dwarfed by
the amount of work dealing with other aspects of the problem.

Moreover, unlike past work which has carved off a slice of coreference (e.g.
the Winograd schema work), this paper makes a big impact on the metrics of the
*overall* coreference problem on a domain (Wikipedia) that many in the ACL
community are pretty interested in.

THE TECHNIQUES: Overall, the techniques are not the strong point of this paper,
though they do seem to be effective. The features seem pretty sensible, but it
seems like additional conjunctions of these may help (and it's unclear whether
the authors did any experimentation in this vein).  The authors should also
state earlier in the work that their primary MC resolution system is a binary
classifier; this is not explicitly stated early enough and the model is left
undefined throughout the description of featurization.

MINOR DETAILS:

Organization: I would perhaps introduce the dataset immediately after "Related
Works" (i.e. have it be the new Section 3) so that concrete results can be
given in "Baselines", further motivating "Approach".

When Section 4 refers to Dcoref and Scoref, you should cite the Stanford papers
or make it clear that it's the Stanford coreference system (many will be
unfamiliar with the Dcoref/Scoref names).

The use of the term "candidate list" was unclear, especially in the following:

"We leverage the hyperlink structure of the article in order to enrich the list
of mentions with shallow semantic attributes. For each link found within the
article under consideration, we look through the candidate list for all
mentions that match the surface string of the link."

Please make it clear that the "candidate list" is the set of mentions in the
article that are possible candidates for being coreferent with the MC.        I think
most readers will understand that this module is supposed to import semantic
information from the link structure of Wikipedia (e.g. if a mention is
hyperlinked to an article that is female in Freebase, that mention is female),
so try to keep the terminology clear.

Section 6.1 says "we consider the union of WCR mentions and all mentions
predicted by the method described in (Raghunathan et al., 2010)." However,
Section 4.1 implies that these are the same? I'm missing where additional WCR
mentions would be extracted.